# Displacement-Invariant Matching Cost Learning for Accurate Optical Flow Estimation

*Jianyuan Wang[1], *Yiran Zhong[1,5], Yuchao Dai[2], Kaihao Zhang[1,4], Pan Ji[3], Hongdong Li[1,5]

[1]Australian National University, [2]Northwestern Polytechnical University,
[3]NEC Labs America, [4]Tencent AI Lab, [5]ACRV

{yiran.zhong, kaihao.zhang, hongdong.li}@anu.edu.au,
jywang.js@gmail.com, daiyuchao@nwpu.edu.cn, panji@nec-labs.com

## Abstract

Learning matching costs has been shown to be critical to the success of the state-of-the-art deep stereo matching methods, in which 3D convolutions are applied on a 4D feature volume to learn a 3D cost volume. However, this mechanism has never been employed for the optical flow task. This is mainly due to the significantly increased search dimension in the case of optical flow computation, *i.e.*, a straightforward extension would require dense 4D convolutions in order to process a 5D feature volume, which is computationally prohibitive. This paper proposes a novel solution that is able to bypass the requirement of building a 5D feature volume while still allowing the network to learn suitable matching costs from data. Our key innovation is to decouple the connection between 2D displacements and learn the matching costs at each 2D displacement hypothesis independently, *i.e.*, displacement-invariant cost learning. Specifically, we apply the same 2D convolution-based matching net independently on each 2D displacement hypothesis to learn a 4D cost volume. Moreover, we propose a displacement-aware projection layer to scale the learned cost volume, which reconsiders the correlation between different displacement candidates and mitigates the multi-modal problem in the learned cost volume. The cost volume is then projected to optical flow estimation through a 2D soft-argmin layer. Extensive experiments show that our approach achieves state-of-the-art accuracy on various datasets, and outperforms all published optical flow methods on the Sintel benchmark. The code is available at `https://github.com/jytime/DICL-Flow`.

## 1   Introduction

Both the optical flow estimation and stereo matching aim to find per-pixel dense correspondences between a pair of input images. In essence, stereo matching can be viewed as a special case of optical flow where the general 2D flow vector search reduces to 1D search along the epipolar lines. Despite this similarity, current leading deep stereo matching methods [15, 44, 41] and leading deep optical flow methods [12, 29, 37] seem to follow very different matching strategies and network architectures. In particular, while both stereo matching and optical flow estimation rely on the cost volume representation, they differ in how to build the cost volumes.

The state-of-the-art deep stereo matching methods [15, 44, 41] learn the matching costs between shifted features of left and right images, whereas most existing deep optical flow methods often rely on non-learned metrics such as dot product [12, 29] and cosine similarity [37]. With the introduction of

learned costs, stereo matching methods can directly obtain disparity maps from the cost volumes with a 1D soft-argmin layer. It has been recognised that learning data-adaptive matching cost is the key to the recent significant advancement of stereo matching methods [15, 44, 41], yet no similar conclusion has been made for the task of deep optical flow. One of the main reasons for such a discrepancy is due to the prohibitive computational cost if one attempts to naively apply the matching cost learning mechanism to optical flow. To learn the matching costs, stereo matching methods only need to construct a 4D feature volume ($2L \times D \times H \times W$, where $L, D, H, W$ denote the feature dimension, disparity range, image height, and image width respectively) by traversally concatenating feature maps between stereo pairs on each disparity shift and compute the 3D cost volume through a series of 3D convolutions. In stark contrast, in optical flow estimation, since the searching space becomes two-dimensional, a direct extension would result in a 5D feature volume ($2L \times U \times V \times H \times W$, where $U, V$ are the 2D search window dimension) and need 4D convolutions to process it, which is computationally very expensive and limited by current computing resources.

In this paper, we propose a novel solution which bypasses dense 4D convolutions and allows the network to learn matching costs from data without constructing a 5D feature volume. By our network design, the matching costs are efficiently learned through a series of *2D convolutions*. Compared with the methods using non-learned metrics, our method achieves a much higher accuracy without obviously sacrificing computational speed. The key idea is to decouple the connections between different 2D displacements in learning the matching costs, which is called displacement-invariant cost learning (DICL). Specifically, we apply the same 2D convolution-based matching net independently on each 2D displacement candidates to form a 4D cost volume ($U \times V \times H \times W$).

Compared with applying 4D convolutions to a 5D feature volume, our proposed matching net has decoupled the connection along 2D displacements, which removes the correlation between different displacement hypothesis. Therefore, we further propose a displacement-aware projection (DAP) layer to scale the learned cost volume along the displacement dimension and mitigate the mutli-modal problem [15] in the learned cost volume. The scaled cost volume is then projected to an optical flow prediction by a 2D soft-argmin layer.

Our contributions are summarized as: 1) To our best knowledge, our method is the first one to learn matching costs from concatenated features for optical flow estimation, through introducing a displacement-invariant cost learning module; 2) We propose a displacement-aware projection layer to reconsider the correlation between different motion hypothesis; and 3) Our method achieves state-of-the-art accuracy on multiple datasets and outperforms all published optical flow estimation methods on the Sintel benchmark. We also provide extensive quantitative and qualitative analysis to verify the effectiveness of our approach.

## 2   Related Work

**Optical Flow Estimation**   Aiming to find per-pixel dense correspondences between a pair of images, optical flow estimation is a fundamental vision problem and has been studied for decades [8]. FlowNet [5] makes the first attempt to use deep learning for optical flow estimation, which directly regresses the optical flow estimation from a pair of images with an encoder-decoder neural network. DCFlow [35] replaces the handcrafted image features with learned feature descriptors and utilizes conventional cost aggregation steps to process the cost volume. Recently, SpyNet [25], PWC-Net [29], and LiteFlowNet [9] combine conventional strategies such as pyramid, warping, and cost volume into network design and achieve impressive performance across different benchmarks. Building upon these methods, Neoral *et al.* [24] estimate the occlusion masks before flow, Hur and Roth [11] utilize a residual manner for iterative refinement with shared weights, and Yin *et al.* [39] hierarchically estimate local matching distributions and compose them together to form a global density. VCN [37] further proposes to construct a multi-channel 4D cost volume and use separable volumetric filtering to avoid the significant amount of memory and computation. SelFlow [20] leverages self-supervised learning on large-scale unlabelled data and uses the well-trained model as an initialization for supervised fine-tuning. Additionally, unsupervised learning of optical flow also witnesses a great progress with the supervision from view synthesis [14]. Researchers further exploit the cues from occlusion reasoning [34, 22], epipolar constraint [43], sequence [13, 7], cross-task consistency [45, 40, 26, 19], knowledge distillation [18, 20], and diverse transformations [17].

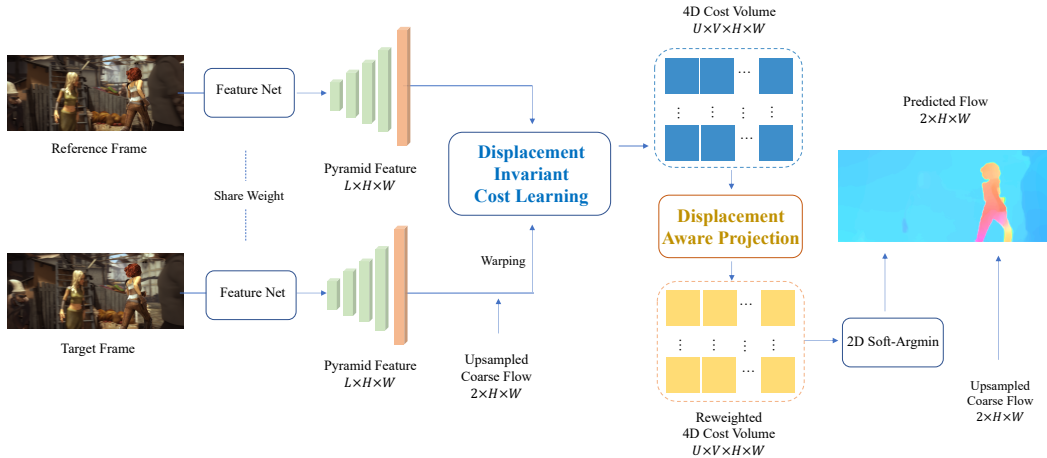

Figure 1: **Overall Architecture and Flow Prediction Process at One Pyramid Level.** The feature net outputs features at five pyramid levels. We take the finest feature level (orange) as an example of flow prediction. The displacement-invariant cost learning module (our matching net) accepts the reference frame feature and warped target frame feature as input, and then outputs a 4D cost volume. The displacement-aware projection layer further reweights the learned cost volume according to per-pixel cost distribution. We then apply 2D soft-argmin on the reweighted costs along the $U$ and $V$ dimensions and hence achieve an optical flow estimation. For clarity, the context network is not visualized here.

Very recently, Zhao *et al.* [42] handle the occlusion caused by feature warping without explicit supervision while Bar-Haim and Wolf [2] focus on sampling difficult examples during training. Different from the traditional coarse-to-fine mechanism, Teed and Deng [32] propose to construct a correlation volume for all pixel pairs and iteratively conduct lookups through a recurrent unit, which shows a significant performance improvement. However, accurate optical flow estimation is still an open and challenging task, especially for small objects with large motions, textureless regions, and occlusion areas.

**Learning Matching Cost** In dense visual matching, the matching cost measures the similarity/dissimilarity between reference frame pixels and the associated target frame pixels. Allowing the network to learn the matching cost from data is a common practice in stereo matching. The first pioneer, MC-CNN-arct [33], utilizes several fully connected layers to compute the similarity score of two patches. GCNet [15] builds a 4D feature volume by concatenating feature maps between stereo pairs on each disparity shift and allows the network to learn matching cost from it with 3D convolutions. It then becomes some kinds of gold standard pipeline in deep stereo matching [44, 41, 4]. However, to the best of our knowledge, the mechanism of learning matching costs with concatenated features has not been employed for optical flow estimation due to the giant 5D feature volume. The closest method VCN [37] leverages an intermediate strategy. Instead of building a 5D volume with concatenated *shifted feature maps*, it constructs a multi-channel 4D cost volume by concatenating 4D cost volumes, where each cost volume stores the cosine similarity between the shifted features. A separable 4D convolution is proposed to process the multi-channel 4D cost volume. Our method, on the other hand, can learn matching costs directly from concatenated features and outperforms VCN with a notable margin across various benchmarks.

## 3 Method

In this section, we first describe the overall architecture and then show how to learn the matching costs without using a 5D feature volume and 4D convolutions. We further provide the design principle of our displacement-aware projection layer.

### 3.1 Overall Architecture

Unlike stereo matching, optical flow methods are often required to handle large 2D displacements. Constructing a full-size cost volume with all possible displacement hypothesis is considered prohibitive in this case. Therefore, following [29], we use a coarse-to-fine warping scheme. As shown in Figure 1, our network adopts a feature net for feature extraction, which consists of five pyramid

levels with $\{1/4, 1/8, 1/16, 1/32, 1/64\}$ resolutions of the input image. At each level, a matching net computes the matching costs with a max displacement of 3 along both the vertical and horizontal directions and a displacement-aware projection layer further scales the learned costs along the displacement plane. Then, a 2D soft-argmin module projects the matching costs to an optical flow estimation. Additionally, we also adopt a context network (similar to [29]) to integrate contextual information and post-process the optical flow estimation with the help of dilated convolution.

## 3.2 Displacement-Invariant Cost Learning Module

Since the cost volumes constructed by non-learned matching costs may omit rich information and limit the ability of subsequent layers, the state-of-the-art deep stereo matching methods leverage 4D feature volumes and 3D convolutions to learn matching costs. However, a direct extension to optical flow will result in a prohibitively 5D feature volumes and 4D convolutions, which is impractical since it will occupy much more GPU memories than 3D convolutions used in stereo. Previous optical flow estimation methods [5, 12, 29] avoids this problem by using a fixed matching cost function (*e.g.*, dot product or cosine similarity) with learned image features. Recently, this problem has been partially addressed by VCN [37]. Instead of storing a full 5D feature volume, they propose a multi-channel cost volume to reduce the size of the channel dimension.

Let $\mathbf{F}^1, \mathbf{F}^2 \in \mathcal{R}^{L \times \lambda H \times \lambda W}$ be the $L$-dimensional feature maps of the $H \times W$ source and target image, which are extracted by the same feature net $f(\cdot)$ with a resolution factor $\lambda$ (*e.g.*, $1/4$) of the input image resolution. We denote a displacement as $\mathbf{u} \in \mathcal{R}^{U \times V}$ and the set of all displacements as $\mathcal{U}$. In our practice the max displacement is 3, and hence $\mathcal{U} = \{(-3, -3), (-3, -2), ..., (0, 0), ..., (3, 3)\}$. VCN computes the matching cost for a pixel $\mathbf{p}$ at a displacement $\mathbf{u}$ by the cosine similarity as:

$$C_{\text{vcn}}(\mathbf{p}, \mathbf{u}) = \frac{\mathbf{F}^1(\mathbf{p}) \cdot \mathbf{F}^2(\mathbf{p} + \mathbf{u})}{||\mathbf{F}^1(\mathbf{p})|| \cdot ||\mathbf{F}^2(\mathbf{p} + \mathbf{u})||}, \tag{1}$$

Then a 4D cost volume $C_{\text{vcn}} \in \mathcal{R}^{U \times V \times \lambda H \times \lambda W}$ is constructed by concatenating the matching cost $C_{\text{vcn}}(\mathbf{p}, \mathbf{u})$ on all displacement candidates in a $U \times V$ window. To form a multi-channel cost volume $C_{\text{vcn}}^K \in \mathcal{R}^{K \times U \times V \times \lambda H \times \lambda W}$, $K$ 4D cost volumes computed from $K$ different features are concatenated, where the volume size is reduced $\frac{2L}{K}$ times compared with the full feature volume.

In contrast, we propose a displacement-invariant cost learning (DICL) module to bypass the requirement of building a 5D feature volume while still allowing the network to learn matching costs with 2D convolutions. Mathematically, for each displacement candidate $\mathbf{u} \in \mathcal{R}^{U \times V}$, we can concatenate features to form a feature map $\mathbf{F_u}$ over all pixels $\mathbf{p} \in \mathcal{R}^{\lambda H \times \lambda W}$:

$$\mathbf{F_u}(\mathbf{p}) = \mathbf{F}^1(\mathbf{p}) \,||\, \mathbf{F}^2(\mathbf{p} + \mathbf{u}), \tag{2}$$

where $||$ concatenates the feature vectors $\mathbf{F}^1(\mathbf{p})$ and $\mathbf{F}^2(\mathbf{p} + \mathbf{u})$. Therefore, for every displacement hypothesis $\mathbf{u}$, $\mathbf{F_u}$ is a concatenated feature map $\in \mathcal{R}^{2L \times \lambda H \times \lambda W}$ specified by $\mathbf{u}$ and can be processed by 2D convolutions. In this paper, we propose to apply a 2D matching net $G(\cdot)$ to learn the cost:

$$C_{\text{ours}}(\mathbf{p}, \mathbf{u}) = [G(\mathbf{F_u})](\mathbf{p}). \tag{3}$$

For every displacement hypothesis $\mathbf{u}$ in the $U \times V$ search window, we apply the same matching net $G(\cdot)$ to learn the matching cost. In other words, instead of processing a 5D feature volume as a whole, we independently process concatenated features $\mathbf{F_u}$ along the displacement dimension through 2D matching net $U \times V$ times. Therefore, we avoid the requirement of storing a 5D feature volume and conducting 4D convolutions. Our method is displacement-invariant cost learning (DICL) as the

Table 1: **Per Layer Analysis of Processing a 5D Feature Volume** ($K \times U \times V \times \lambda H \times \lambda W$), where $K$ denotes the number of different features in VCN and the dimension of concatenated features in ours. It may be noted that the gradients need to be stored in full grid for back propagation during the training phase.

| Methods | Kernel | Params | ratio | Theoretical Inference Memory | ratio |
|---------|--------|--------|-------|------------------------------|-------|
| 4D conv. | $(K, K, 3, 3, 3, 3)$ | $81\,K^2$ | $9K$ | $K \times U \times V \times \lambda H \times \lambda W$ | $U \times V$ |
| VCN | $(2, K, K, 3, 3)$ | $18\,K^2$ | $2K$ | $K \times U \times V \times \lambda H \times \lambda W$ | $U \times V$ |
| Ours | $(K, 3, 3)$ | $9\,K$ | $1$ | $K \times \lambda H \times \lambda W$ | $1$ |

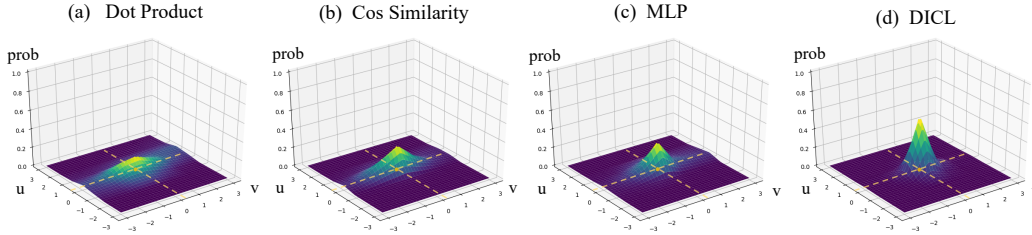

Figure 2: **Qualitative Example of the Displacement Probability Distribution with Different Kinds of Matching Costs.** The intersection of two yellow lines shows the ground truth location. 'MLP' indicates predicting the matching cost with a three-layer multilayer perceptron.

learned matching cost only depends on the current displacement shift $\mathbf{u}$. It is worth noting that our method also supports a parallel implementation of the matching net for different displacements.

Our approach allows the network to learn matching cost fully from data without using any non-learned metrics. Non-learned metrics such as dot product and cosine similarity generally cannot exploit the rich information of high-dimensional features, and constructing an appropriate distance metric itself has been a non-trivial research topic [38, 16]. A qualitative example is shown in Figure 2 where the DICL module achieves a sharper displacement probability (costs after softmax) distribution than other methods. The probabilities of our result closely gather around the ground truth displacement. Moreover, our method does not need to construct a 5D volume and can efficiently compute the matching cost by 2D convolutions. A detailed analysis of VCN and our method in processing a 5D volume ($K \times U \times V \times \lambda H \times \lambda W$) is shown in Table 1. For each layer, our method requires $\frac{1}{2K}$ trainable parameters and $\frac{1}{U \times V}$ memory consumption of the VCN. In real case, VCN requires 1.9G memory for a pair of images with a crop size of [256, 384] in training while ours only need 1.1G. It is worth noting that ours is $55.56\%$ faster in inference (0.08s *vs.* 0.18s on Chairs dataset).

### 3.3 Displacement-Aware Projection Layer

Under our network, the DICL module decouples the connections among the displacement dimension, where the same 2D convolution based matching net is applied on each displacement hypothesis independently. However, we may to some extent weaken the correlation between displacement candidates, compared with directly applying 4D convolutions to a 5D feature volume. To remedy this issue, we propose to reweight the matching cost at each displacement plane by a linear combination of the matching costs at all the displacements. Specifically, for each pixel $\mathbf{p}$, denote a displacement candidate $\mathbf{u} \in \mathcal{R}^{U \times V}$ as $(u, v)$, and then the corresponding $C_{\mathbf{u}}$ denotes the pixel's matching cost at $\mathbf{u} = (u, v)$. The new weighted matching cost $C'_{\mathbf{u}}$ is obtained as:

$$C'_{\mathbf{u}} = \sum_{\mathbf{v} \in \mathcal{U}} w_{(\mathbf{u}, \mathbf{v})} C_{\mathbf{v}}, \tag{4}$$

where $w_{(\mathbf{u}, \mathbf{v})}$ denotes the learned reweighting parameters between the displacement $\mathbf{u}$ and $\mathbf{v}$, and $\mathcal{U}$ is the set of all displacements. We term this as a Displacement-Aware Projection (DAP) Layer. Along the displacement dimension direction, each slice of the matching cost volume shares the same displacement hypothesis, *i.e.*, 2D optical flow (motion) vector. Our proposed DAP layer exploits the correlation between different displacement candidates to achieve better matching cost estimation, and is thus displacement-aware.

Our DAP layer is implemented as a $1 \times 1$ 2D convolution layer as Eq. 4 is defined invariant to the pixel location. We first reshape the 4D cost volume $\mathcal{R}^{U \times V \times \lambda H \times \lambda W}$ to a 3D cost volume $\mathcal{R}^{N \times \lambda H \times \lambda W}$, where $N = U \times V$. Then the DAP layer is applied to the 3D cost volume to adaptively adjust the cost distributions along the $N$ dimension. After that, the 3D cost volume is reshaped back for the following 2D soft-argmin operation. We visually show the working mechanism of the DAP layer in Figure 3, and the detailed analysis is in Section 4.3.

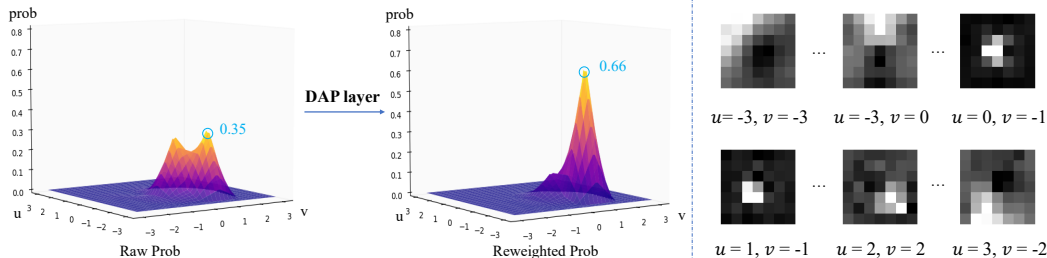

Figure 3: **Multi-Modal Effect and Visualization of the DAP Layer Kernels.** The left column compares an example pixel's displacement probability, before and after using DAP layer. The right column visualizes several kernels from a well-trained DAP layer, where white indicates a high value. The DAP layer has $U \times V$ kernels in total and each kernel maps the input costs to a specific $u$ and $v$. The $u$ and $v$ below each kernel indicate the corresponding displacement hypothesis of its output cost. The qualitative analysis is provided in Section 4.3.

## 3.4 Matching Cost to Optical Flow

To achieve a flow from a learned cost volume, a traditional way is to extract the displacement with the highest probability/lowest matching cost with the Winner-Takes-All (WTA) strategy, *i.e.*, argmin operation. However, due to the discrete nature of the cost volume, such a process will not be able to recover sub-pixel level estimation and is non-differentiable. In stereo matching, a common strategy is to use a soft-argmin [15] operator to make it differentiable and is able to achieve sub-pixel accuracy. Hence, similar to [37], we extend the 1D soft-argmin operation used in disparity estimation to a 2D soft-argmin operation to project the 4D cost volume into an optical flow prediction as below,

$$\hat{\mathbf{u}} = \sum_{\mathbf{u} \in \mathcal{U}} [\mathbf{u} \times p(\mathbf{u}))], \tag{5}$$

where $\mathbf{u}$ denotes the displacement vector $(u, v)$, $p(\mathbf{u}) = \sigma(-C'_{\mathbf{u}})$ is the probability of displacement $\mathbf{u}$ after 2D softmax operation along the displacement hypothesis space. Therefore, $\hat{\mathbf{u}}$ is the expected optical flow vector.

However, the soft-argmin operation is affected by probabilities of all displacement candidates, which makes it susceptible to multi-modal distributions. That means, it will select the mean value of various peaks rather than the highest one. The multi-modal distributions are common in occlusion, repetitive, textureless, and blurry areas. Statistically, we have found our displacement-aware projection layer can largely mitigate this problem. We defer this discussion to the following section.

## 4 Experiments

In this section, we provide implementation details and training strategies of our network. We compare our results with existing state-of-the-art optical flow methods on standard optical flow benchmarks, *i.e.*, KITTI 2015 [23] and Sintel [3], and provide both quantitative and qualitative results. We also analyze the effect of different matching cost metrics as well as our proposed Displacement-Invariant Cost Learning (DICL) module and Displacement-Aware Projection (DAP) layer.

### 4.1 Implementation Detail

**Network Architecture**  To avoid redundant computation, our feature net projects all the pyramid level features into 32 dimensions. Therefore the input feature map $\mathbf{F_u}$ of a matching net $G(\cdot)$ would have 64 dimensions. We denote a 2D convolution layer specification as $[C_{in}, C_{out}, kernel\ size, stride]$. Our matching net consists of six 2D convolution layers, whose specifications sequentially follow $[64, 96, 3, 1]$, $[96, 128, 3, 2]$, $[128, 128, 3, 1]$, $[128, 64, 3, 1]$, $[64, 32, 4, 2]$, $[32, 1, 3, 1]$. The second layer downsamples the feature map and the fifth layer is a deconvolution layer which upsamples the feature back. All layers except the last one adopt ReLU and batch normalization. We adopt level-specific matching nets at different pyramid levels. The displacement-aware projection layer is implemented as a $1 \times 1$ convolution along the displacement dimension with a feature dimension of 49 ($U \times V$, and $U = V = 7$).

Table 2: **Quantitative Results on KITTI 2015 and Sintel Datasets.** The metric EPE is the average endpoint error and Fl-all is the percentage of erroneous pixels over all pixels. The symbol 'C+T' indicates a model pre-trained on the Chair and Things datasets while '+K/S' means further fine-tuned on the KITTI or Sintel dataset. Parentheses means the results are reported on its training dataset. The unavailable results are marked as '-'. Best results of 'C+T' and '+K/S' models are separately bolded. Ours-w/o DAP is the model without displacement-aware projection layer. The time setting refers to [29] and is reported on an NVIDIA 1080Ti GPU.

| | Method | Time | K-15 train | | K-15 test | S-train (EPE) | | S-test (EPE) | |
|---|---|---|---|---|---|---|---|---|---|
| | | (s) | EPE | Fl-all | Fl-all | Clean | Final | Clean | Final |
| | EpicFlow [27] | 15.00 | - | - | 26.29 | - | - | 4.12 | 6.29 |
| | DCFlow [35] | 8.60 | - | 15.1 | 14.86 | - | - | 3.54 | 5.12 |
| C+T | FlowNet2 [12] | 0.12 | 10.08 | 30.0 | - | 2.02 | **3.54** | 3.96 | 6.02 |
| | PWCNet [29] | 0.03 | 10.35 | 33.7 | - | 2.55 | 3.93 | - | - |
| | LiteFlowNet [9] | 0.09 | 10.39 | 28.5 | - | 2.48 | 4.04 | - | - |
| | LiteFlowNet2 [10] | 0.04 | 8.97 | 25.9 | - | 2.24 | 3.78 | - | - |
| | HD$^3$F [39] | 0.08 | 13.17 | 24.0 | - | 3.84 | 8.77 | - | - |
| | VCN [37] | 0.18 | **8.36** | 25.1 | - | 2.21 | 3.62 | - | - |
| | Ours-w/o DAP | 0.08 | 8.78 | 23.8 | - | 2.11 | 3.85 | - | - |
| | Ours | 0.08 | 8.70 | **23.6** | - | **1.94** | 3.77 | - | - |
| +K/S | FlowNet2 [12] | 0.12 | (2.30) | (8.6) | 11.48 | (1.45) | (2.01) | 4.16 | 5.74 |
| | PWCNet+ [30] | 0.03 | (1.50) | (5.3) | 7.72 | (1.71) | (2.34) | 3.45 | 4.60 |
| | LiteFlowNet [9] | 0.09 | (1.62) | (5.6) | 9.38 | (1.35) | (1.78) | 4.54 | 5.38 |
| | LiteFlowNet2 [10] | 0.04 | (1.47) | (4.8) | 7.74 | (1.30) | (1.62) | 3.45 | 4.90 |
| | IRR-PWC [11] | 0.21 | (1.63) | (5.3) | 7.65 | (1.92) | (2.51) | 3.84 | 4.58 |
| | HD$^3$F [39] | 0.08 | (1.31) | (4.1) | 6.55 | (1.87) | **(1.17)** | 4.79 | 4.67 |
| | SelFlow [20] | 0.09 | (1.18) | - | 8.42 | (1.68) | (1.77) | 3.74 | 4.26 |
| | VCN [37] | 0.18 | (1.16) | (4.1) | **6.30** | (1.66) | (2.24) | 2.81 | 4.40 |
| | Ours-w/o DAP | 0.08 | (1.09) | (3.8) | - | (1.30) | (1.72) | - | - |
| | Ours | 0.08 | **(1.02)** | **(3.6)** | 6.31 | **(1.11)** | (1.60) | **2.12** | **3.44** |

**Training** Similar to [12, 29], we first pre-train the network on the synthetic dataset FlyingChair [5] for 150K iterations. The learning rate ($lr$) is initially $0.001$ and reduced by half after 120K. The model is further trained on FlyingThings [21] for 220K iterations with a $lr$ of $0.00025$. Then, we separately fine-tune the network on the Sintel [3] and KITTI [23] dataset for 60K iterations, starting from a $lr$ of $0.00025$ and dropping it by half at 30K, 50K. We use a batch size of $8$ per GPU on the FlyingChair and $2$ on other datasets, with $8$ NVIDIA 1080 Ti GPUs for training.

We use a multi-level $\ell_2$ loss for training on all datasets and the loss weights are $1.0, 0.75, 0.5, 0.5, 0.5$ for flows predicted from $1/4$ to $1/64$ resolution. We also follow the augmentation strategy of [37], including random resizing, cropping, flipping, color jittering, and asymmetric occlusion. To avoid noisy training in the initial stage, only random cropping is applied in the first 50K training on the FlyingChair. More details are available in the supplementary material.

## 4.2 Quantitative Result

**Benchmark Result** As shown in Table 2, our approach shows its superiority on various datasets. On the Sintel 'final' pass benchmark, we achieve an average endpoint error (EPE) of $3.44$, which is significantly better than previous state-of-the-art VCN ($4.40$) and SelFlow ($4.26$). Our approach also outperforms all the previous methods on the 'clean' pass with an EPE of $2.12$. With the help of the DICL module, our inference speed is even faster although we predict matching costs in a learning-based manner. Additionally, if not fine-tuned (see the C+T part of Table 2), our method achieves a clear performance improvement on the 'clean' pass as well, whereas the only comparable method FlowNet2 [12] uses over $1.5\times$ inference time. On the KITTI 2015 training set, our approach reaches the smallest EPE and lowest error percentage Fl-all. For the testing set, it is close to the best one. It is worth noting that KITTI 2015 provides limited training samples (merely 200) which may constrain our learned costs. Instead, if not fine-tuned on KITTI training set (only using Chair and Thing), our method obtains the best Fl-all among all approaches, which demonstrates its performance when sufficient data is provided and its strong generalization ability.

Table 3: **Ablation study on cost computation metrics.** The models for 'Chair' were trained on the Chairs dataset. The models for 'K-15' and 'S' were trained on Things dataset.

| Method | Chairs | KITTI-15 train | | Sintel-train (EPE) | |
|---|---|---|---|---|---|
| | EPE | EPE | Fl-all | Clean | Final |
| Dot Product | 1.86 | 10.39 | 31.1 | 2.57 | 4.06 |
| Cosine Similarity | 1.84 | 10.45 | 30.2 | 2.55 | 4.03 |
| 3-Layer MLP | 1.76 | 9.83 | 28.9 | 2.45 | 3.98 |
| Reduced DICL | 1.72 | 9.77 | 28.3 | 2.42 | 3.99 |
| DICL | 1.33 | 8.78 | 23.8 | 2.11 | 3.85 |

**Displacement-Invariant Cost Learning** As shown in Table 3, we also conduct an ablation study to explore the superiority of our DICL module. We compare different ways of computing the matching cost by dot product [12, 29], cosine similarity [37], a three-layer MLP, reduced DICL (the same network structure as DICL but with $1 \times 1$ convolution kernels), and our DICL module. The DICL module significantly outperforms all other methods. On the FlyingChair validation dataset, the dot product and cosine similarity share a close performance ($1.86$ and $1.84$). The three-layer multilayer perceptron (MLP) and the reduced DICL module take pixel-to-pixel concatenated features as input and learns the distribution of matching cost, hence achieve a slight improvement ($1.76$ and $1.72$). On the contrary, in learning the matching costs, our DICL module not only considers the similarity between features $\mathbf{F_u(p)}$ and $\mathbf{F_u(p + u)}$ but also takes the spatial context into consideration via 2D convolutions. Compared with using MLP and reduced DICL, the DICL module provides extra spatial context information and hence successfully utilizes the power of matching cost learning. The DICL module improves the EPE from around $1.8$ to $1.33$. Their performance on the Sintel and KITTI dataset show the same trend. A qualitative example is provided in Figure 2.

To further prove the validity of our DICL module, we replace the non-learned metrics of two well-known pipelines PWCNet [29] and VCN [37] with our DICL module and report the results on the Chairs dataset in Table 4.

Table 4: **PWCNet and VCN with our DICL module.** Models were trained and evaluated on the Chairs dataset.

| Method | PWCNet | PWCNet+DICL | VCN | VCN+DICL |
|---|---|---|---|---|
| Chair EPE | 2.00 | 1.83 | 1.66 | 1.45 |

With our DICL module, both PWCNet and VCN achieve a notable improvement: 8.5% for PWCNet ($2.00$ *vs.* $1.83$) and 12.7% for VCN ($1.66$ *vs.* $1.45$).

**Displacement-Aware Projection Layer** As reported in Table 2, although the counterpart 'Ours-w/o DAP' model has showed outstanding performance, the DAP layer generally improves its EPE by around 0.1 pixel on various datasets. The improvement is consistent no matter if fine-tuning on specific datasets, which shows the correlation of motions is not limited to datasets and verifies the generalization ability of the DAP layer.

## 4.3 Qualitative Analysis

**Multi-Modal Effect and How the DAP Layer Works** To explore how the DAP layer improves accuracy, we visualize the probability maps before and after the DAP layer. A typical example is shown in Figure 3. The initial probability map has two peaks with the highest probability of $0.35$. Although these two peaks are spatially close, it constrains the network's confidence on sub-pixel prediction. As discussed in Section 3.4, such a multi-modal problem is an inherent drawback of the soft-argmin operation. The DICL module cannot solve this problem because it decouples the feature connections along the displacement ($UV$) dimension. Instead, the

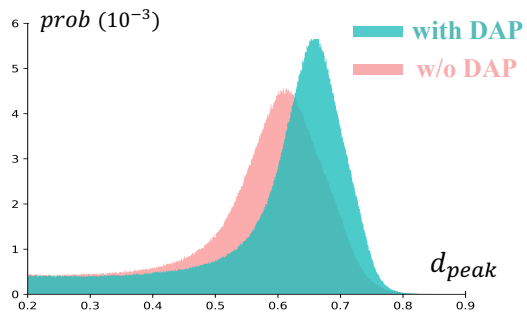

Figure 4: **Histogram of the $d_{peak}$ distribution with and without the DAP layer**, with a bin size of 0.001.

DAP layer re-weights the costs of different displacements, dynamically selects the optimal one, and

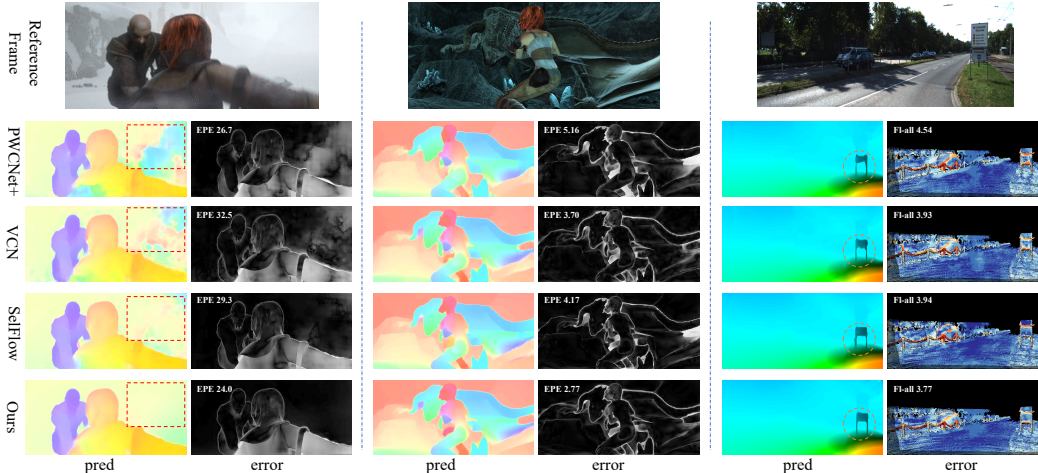

Figure 5: **Qualitative Results on the Sintel and KITTI 2015 Test Dataset.** The results are downloaded from the benchmark websites. The left and middle examples come from Sintel while the right one is from KITTI. For each example, the left column compares the predicted flow and the right column provides the error map against ground truth flow. The white metric value on the left top corner indicates the EPE or Fl-all of this example.

hence improves the prediction confidence (now the highest probability is 0.66). Statistically, we can depict the degree of the multi-modal problem as $d_{peak}$, the difference value between the highest value and the second of the $U \times V$ probability map. On the Sintel dataset, the DAP layer increases the median $d_{peak}$ from 0.57 to 0.63, which is an obvious improvement since $d_{peak} \in [0, 1]$. The detailed distribution of $d_{peak}$ is shown in Figure 4.

To further analyze how the DAP layer solves this problem, we also visualize its kernels as shown in Figure 3. The DAP layer is a $1 \times 1$ convolution with $U \times V$ kernels. Each kernel maps the $U \times V$ costs (49 in our implementation) to a new one. It can be verified that, a kernel tends to have a high reaction to input cost $C_{in}(u, v)$ and its neighbour displacement candidates, if this kernel maps input costs to $C_{out}(u, v)$. This conforms to our design target that displacement candidates belonging to the same motion (close in the $UV$ space) should share close confidence, since motions tend to be continuous. The learned kernels also have negative values (generally the dark regions in kernel visualization), which indicates a peak would be suppressed if another far-away peak exists.

**Visualization of Benchmark Result**   We compare our visualization results on the Sintel and KITTI 2015 benchmarks, as shown in Figure 5. It is worth noting that for the textureless region (indicted by a red rectangle on the left example), our method provides a confident and smooth prediction while the results of others are noisy, which shows the benefit of matching cost learning. The middle example verifies that our method can handle complicated objects and fast motions, where we achieve a much lower EPE than the competing methods. The right example comes from the KITTI benchmark and indicates that our approach is also good at small objects. The leg of the road sign (circled by orange) is very thin and only occupies several pixels. Our result accurately depicts the road sign's contour while those of other methods are twisted.

## 5   Conclusion

To the best of our knowledge, we are the first one to learn the matching costs from concatenated features for optical flow estimation. To overcome the necessity of building a 5D feature volume and conducting 4D convolutions, we decouple the connections between different displacements in computing the matching costs, which enables us to apply the same 2D convolution based matching net to each displacement independently. To further handle the multi-modal issue in the learned cost volume, we introduce a displacement-aware projection layer, which scales the matching costs by exploiting the correlation among different displacements. Experimental results on the KITTI 2015 and Sintel datasets show that our method obtains a new state-of-the-art. In the future, we plan to further speed up the implementation to achieve real-time performance.

## Broader Impact

This paper proposes to predict accurate optical flows via matching cost learning. With the rapid development of optical flow estimation, we have seen its wide applications on autonomous driving [6, 23], medical analysis [28, 36], human motion recognition [1, 31], and so on. These downstream tasks are closely associated with people's lives and some of them (*e.g.*, autonomous driving) may reshape the division of labour in our society. Our proposed efficient and accurate flow estimation method can be generally applied in these fields and bring more convenience.

However, we should also see the risks besides the positive impacts. First, the development of new technologies may take away some people's jobs and threaten the related families' living. Additionally, reliance on optical flow methods possibly results in potential safety hazards, since current optical flow estimation methods may fail in challenging cases especially for occlusion areas or fast motions. Moreover, the risk of the inaccurate usage also increases.

Unfortunately, the risks mentioned above are mainly raised from the usage of technologies and cannot be solved by the academic community only. Machine learning researchers should carefully consider the abuse of their algorithms and advocate for related social research. The involvement of policy makers and companies is necessary to avoid the risks above.

## Acknowledgments and Disclosure of Funding

Yuchao Dai's research was supported in part by Natural Science Foundation of China (61871325, 61671387) and National Key Research and Development Program of China under Grant 2018AAA0102803. Hongdong Li's research was supported in part by the ARC Centre of Excellence for Robotics Vision (CE140100016) AND ARC-Discovery (DP 190102261), ARC-LIEF (190100080) grants.

## Footnotes

* Indicates equal contribution

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
