[Supplementary Material]

# Displacement-Invariant Matching Cost Learning for Accurate Optical Flow Estimation – Supplementary Material

*Jianyuan Wang[1], *Yiran Zhong[1,5], Yuchao Dai[2], Kaihao Zhang[1,4], Pan Ji[3], Hongdong Li[1,5]
[1]Australian National University, [2]Northwestern Polytechnical University,
[3]NEC Labs America, [4]Tencent AI Lab, [5]ACRV
{yiran.zhong, kaihao.zhang, hongdong.li}@anu.edu.au,
jywang.js@gmail.com, daiyuchao@nwpu.edu.cn, panji@nec-labs.com

## Abstract

In this supplementary material, we provide (a) more implementation details, (b) robustness analysis of our proposed method, and (c) our qualitative results on the FlyingChairs and FlyingThings datasets.

## 1 More Implementation Details

**Datasets** The datasets used in our work include FlyingChairs [3], FlyingThings [6], KITTI 2015 [7], and MPI Sintel [2]. The FlyingChairs dataset is generated by segmented images of chairs and random background images, with $22,232$ image pairs for training and $640$ image pairs for validation. The FlyingThings dataset is also a synthetic dataset but much closer to real-world scenes, including $21,818$ training and $4,248$ validation pairs. The KITTI 2015 dataset includes 200 image pairs for training and 200 pairs for testing. Its ground truth labels are marked by accurate 3D CAD models and hence the number of samples is limited. The Sintel dataset provides naturalistic image pairs derived from an open-resource film (with 3D annotation), with $1041$ training samples and $552$ testing samples. During training, we crop the input images to a size of $[256, 384]$, $[384, 768]$, $[256, 1024]$, and $[384, 768]$ separately on the FlyingChairs, FlyingThings, KITTI 2015, and Sintel datasets.

**Network Architecture** Our network consists of a feature net, a matching net, a motion-aware projection layer, a 2D soft-argmin module, and a context network. We adopt the feature extraction module of GANet [11] as our feature net with some modifications: we employ five pyramid levels and project the features of each pyramid level into 32 dimensions (dims) by a $3 \times 3$ convolution layer. The structure of the matching net and motion-aware projection layer has been provided in the main paper Section 4.1. The context network takes the predicted flow (2 dims), the entropy of the flow[1] (1 dim), the corresponding level pyramid feature (32 dims), and the resized input image (3 dims) as the input. We show the context network structure of the $1/4$ resolution level in Table 1.

## 2 Robustness Analysis

Recently it has been found that adversarial attacks can fool deep neural networks with just a small patch, *i.e.*, several pixels can mislead the whole prediction result. To analyze the robustness of our proposed method, we perform an adversarial attack on our network. We use the white-box attack method proposed in [9] and try to learn a small patch to fool our network. An example of the learned

Table 1: **The layer parameter description of the context network.** The first two columns indicate the input and output dimension of this layer. The kernel size is the size of the convolution kernel, padding is the shape of zero-padding added to the input, and dilation is the dilation rate. The ReLU, BN, and Bias columns describe if the layer adopt these componenets.

| Input Dim | Output Dim | Kernel Size | Padding | Dilation | ReLU | BN | Bias |
|---|---|---|---|---|---|---|---|
| 38 | 64 | 3 | 1 | 1 | Yes | Yes | No |
| 64 | 128 | 3 | 2 | 2 | Yes | Yes | No |
| 128 | 128 | 3 | 4 | 4 | Yes | Yes | No |
| 128 | 96 | 3 | 8 | 8 | Yes | Yes | No |
| 96 | 64 | 3 | 16 | 16 | Yes | Yes | No |
| 64 | 32 | 3 | 1 | 1 | Yes | Yes | No |
| 32 | 2 | 3 | 1 | 1 | No | No | Yes |

Table 2: **Optical Flow Methods' Performance Against Adversarial Attacks.** The patch size used by the adversarial attack is indicated by pixels, *e.g.*, $25 \times 25$. The column 'Diff' denotes the relative EPE difference after attacks. The results are reported on the KITTI 2015 training set.

| Network | Unattacked EPE | 25x25 EPE | 25x25 Diff | 51x51 EPE | 51x51 Diff | 102x102 EPE | 102x102 Diff | 153x153 EPE | 153x153 Diff |
|---|---|---|---|---|---|---|---|---|---|
| FlowNetC [3] | 14.56 | 29.07 | +14.51 | 40.27 | +25.51 | 82.41 | +67.85 | 95.32 | +80.76 |
| FlowNet2 [4] | 11.90 | 17.04 | +5.14 | 24.42 | +12.52 | 38.57 | +26.67 | 59.58 | +47.68 |
| SpyNet [8] | 20.26 | 20.59 | +0.33 | 21.00 | +0.74 | 21.22 | +0.96 | 21.00 | +0.74 |
| PWC-Net [10] | 11.03 | 11.37 | +0.34 | 11.50 | +0.47 | 11.86 | +0.83 | 12.52 | +1.49 |
| Back2Future [5] | 17.49 | 18.04 | +0.55 | 18.24 | +0.75 | 18.73 | +1.24 | 18.43 | +0.94 |
| Ours | 8.98 | 9.17 | +0.19 | 9.30 | +0.32 | 9.52 | +0.54 | 9.61 | +0.63 |

patch (with a diameter of 51 pixels) is shown in Figure 1. Inside the patch, the network is confused about the motion in this region. However, its prediction on other areas is still confident and not fooled by the adversarial example. We also provide a quantitative robustness analysis in Table 2. The results of other methods come from [9], using the same attacking method. Our approach achieves the lowest EPE no matter if adversarial examples are utilized, and the relative EPE difference is smaller than all other methods, which demonstrates the robustness of our method.

| Attacked Reference Frame | Unattacked Flow | Attacked Flow |
|---|---|---|

Figure 1: **A Qualitative Example of the Adversarial Attack.** Following the setting of [9], a circular patch is learned and added to the image. Our method is robust to the white-box attack, *i.e.*, the neighbouring regions are not fooled by the added patch. Since there is no motion within the patch, our method predicts zero motions.

## 3   Qualitative Examples

We provide the qualitative examples of FlyingChairs and FlyingThings datasets in Figure 2. Our method could predict accurate optical flows although the input image pairs may be noisy (especially the examples from FlyingChairs) and unrealistic. The ground truth and predicted flows are visualized following the Middlebury color style [1].

## Footnotes

[1]We define the entropy of each pixel as: $h = -\sum_{\mathbf{u} \in \mathcal{U}} P(\mathbf{u}) \log(P(\mathbf{u}))$, where $P(\mathbf{u})$ represents the probability of displacement $\mathbf{u}$.

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

| Reference Frame | Ground Truth | Prediction |
| --- | --- | --- |

Figure 2: **Qualitative Results on the FlyingChairs and FlyingThings dataset.** The top two rows come from the FlyingChairs validation set while the bottom two are from the FlyingThings validation set.

[3] Alexey Dosovitskiy, Philipp Fischer, Eddy Ilg, Philip Hausser, Caner Hazirbas, Vladimir Golkov, Patrick Van Der Smagt, Daniel Cremers, and Thomas Brox. Flownet: Learning optical flow with convolutional networks. In *Proc. IEEE Int. Conf. Comp. Vis.*, pages 2758–2766, 2015.

[4] Eddy Ilg, Nikolaus Mayer, Tonmoy Saikia, Margret Keuper, Alexey Dosovitskiy, and Thomas Brox. Flownet 2.0: Evolution of optical flow estimation with deep networks. In *Proc. IEEE Conf. Comp. Vis. Patt. Recogn.*, pages 2462–2470, 2017.

[5] Joel Janai, Fatma Güney, Anurag Ranjan, Michael J. Black, and Andreas Geiger. Unsupervised learning of multi-frame optical flow with occlusions. In Vittorio Ferrari, Martial Hebert, Cristian Sminchisescu, and Yair Weiss, editors, *Computer Vision - ECCV 2018 - 15th European Conference, Munich, Germany, September 8-14, 2018, Proceedings, Part XVI*, volume 11220 of *Lecture Notes in Computer Science*, pages 713–731. Springer, 2018.

[6] Nikolaus Mayer, Eddy Ilg, Philip Hausser, Philipp Fischer, Daniel Cremers, Alexey Dosovitskiy, and Thomas Brox. A large dataset to train convolutional networks for disparity, optical flow, and scene flow estimation. In *Proc. IEEE Conf. Comp. Vis. Patt. Recogn.*, pages 4040–4048, 2016.

[7] Moritz Menze and Andreas Geiger. Object scene flow for autonomous vehicles. In *Proc. IEEE Conf. Comp. Vis. Patt. Recogn.*, June 2015.

[8] Anurag Ranjan and Michael J. Black. Optical flow estimation using a spatial pyramid network. In *Proc. IEEE Conf. Comp. Vis. Patt. Recogn.*, pages 2720–2729, 2017.

[9] Anurag Ranjan, Joel Janai, Andreas Geiger, and Michael J. Black. Attacking optical flow. In *2019 IEEE/CVF International Conference on Computer Vision, ICCV 2019, Seoul, Korea (South), October 27 - November 2, 2019*, pages 2404–2413. IEEE, 2019.

[10] Deqing Sun, Xiaodong Yang, Ming-Yu Liu, and Jan Kautz. PWC-Net: CNNs for optical flow using pyramid, warping, and cost volume. In *Proc. IEEE Conf. Comp. Vis. Patt. Recogn.*, 2018.

[11] Feihu Zhang, Victor Adrian Prisacariu, Ruigang Yang, and Philip H. S. Torr. Ga-net: Guided aggregation net for end-to-end stereo matching. In *IEEE Conference on Computer Vision and Pattern Recognition,*

*CVPR 2019, Long Beach, CA, USA, June 16-20, 2019*, pages 185–194. Computer Vision Foundation / IEEE, 2019.