[Reviews · NeurIPS 2020]

Review 1

Summary and Contributions: This paper addresses the topic of learning a matching costs for optical flow estimation (in contrast to prior work, which learns feature transformations and uses e.g. a cosine loss on top of those). In particular, it addresses the computational issue of convolving the (feature_dim x U x V x H x W) cost volume by factoring it into (a) a convolution taking into account the spatial context, and (b) a convolution taking into account the displacement context. Specifically, for a flow candidate u at location x the features F_1(x) and F_2(x+u) are concatenated (for the full feature maps) and used as input to a cost network, which produces a single cost value at each location x. In a subsequent step, the costs for a set of flow candidates at a given pixel are stacked and re-weighted using a 1x1 convolution, effectively re-weighting the costs for flow candidates at a given pixel based on the previously computed costs. A soft argmin is then used to obtain subpixel accurate flow. The methods achieves state-of-the-art results on Sintel and KITTI.

Strengths: The method proposed a very reasonable alternative of the computationally expensive convolution of the full feature volume to compute the cost volume, by factoring it out into a convolution step across the spatial context (the first, DICL step) and a re-weighting based on the displacement context (the second, MAP step). The results, both quantitatively and qualitatively, are convincing. The ablation studies provide a good amount of additional information, in particular on the reduction of multimodality of the costs after the first step. Lastly, the paper is generally well written and straightforward to understand.

Weaknesses: l. 243 states that the advantage of the proposed DICL over a simple MLP is that DICL takes the context into account, while MLP does not. While this is true, the MLP also seems like a much smaller network. If would therefore be a fairer comparison to compare DICL to a "reduced DICL" network with the same architecture, but with all convolutions of size 1x1. Furthermore, there are a few correctness and clarity issues, which are however minor.

Correctness: The paper is largely correct, except for two minor issues: (1) l. 156 "weaken the correlation" between candidates: Since the proposed methods only processes one displacement value at a time, it does not merely weaken the correlation, but in fact remove it. (2) l. 139 "shuffling of the feature maps across displacement dimension": While shuffling the feature maps would not change the cost volume, it would nevertheless cause a corresponding shuffling of the cost volume (also in displacement direction) -- the cost volume is not invariant to a shuffling of feature maps, but equivariant.

Clarity: The paper is generally well written, apart from the occasional "Besides," at the beginning of sentences. The only exception to this is the use of the term MAP, which is commonly understood as "Maximum A-Posteriori" and not "Motion-Aware Projection". I believe it would greatly reduce friction and mental effort for the reader (especially those coming from the machine learning community) if the authors could come up with a different acronym for their cost volume re-weighting.

Relation to Prior Work: This paper adequately addresses prior work.

Reproducibility: Yes

Additional Feedback: This paper proposes a reasonable decomposition of the computational bottleneck for the application of cost learning for optical flow estimation. The results are good, and I believe that this (or similar) methods could have an impact for the general case of computing cost volumes for high-dimensional variables. I therefore recommend to accept this paper to Neurips. --- Post rebuttal --- My questions were sufficiently addressed in the rebuttal. It is interesting to see that the improvement over the simple MLP seem to come mostly from the larger capacity; the author should probably address this in the camera ready. As a suggestion - since the MAP layer effectively re-weights the costs at a given input pixel, it might be interesting to provide it not just the input costs, but also the reference image context itself, which might help to take structures like image boundaries into account.


Review 2

Summary and Contributions: This paper describes an approach to learn the matching cost for optical flow. It is heavily inspired by current practice in stereo matching and proposes to build a cost volume with the learned cost and features. To circumvent the otherwise the large computational requirements that this would incur in optical flow, the paper proposes to independently (and potentially in parallel) apply the learned cost to the feature that corresponds to each possible displacement. Results are shown on standard benchmarks and indicate good performance with respect to the state-of-the-art.

Strengths: The proposed approach shows good results and is clearly among state-of-the-art methods. Cost volume-based approaches are seeing a ressurgence in optical flow since they seem to provide strong results. The proposed approach fits well within these frameworks. The main contribution is replacing the commonly used cosine similarity with a learned cost that leverages a natural factorization of the problem, while it can still pull spatial information into the computation using the 2D convolutions. The formulation overall makes sense and seems to be generally applicable.

Weaknesses: Cost volumes are a common part of many existing approaches (PWCNet, FlowNet1+2, VCN...) All of these use non-learned metrics to construct the cost from the features. It would have been nice to see if DICL is able to consistently improve results when dropping it into these existing pipelines. This would lend further credibility to the approach as a more general module that is widely applicable. It would also better ablate the factors that lead to the strong performance as any gains would be clearly pinpointed to DICL and not to other choices in the training pipeline. While 5D processing with large correlation windows is clearly infeasible. It is not clear to me if full 5D processing really is a problem in the proposed setting. Displacements are only evaluated over a 3x3 window. Large displacements are handled using the usual coarse-to-fine techniques. This induces tiny cost volumes that are even by an order of magnitude smaller than what is commonly encountered in stereo matching. The problematic scaling behavior of cost volumes in the context of optical flow comes from the size of the correlation window, which quickly induces huge memory and computational requirements due to its quadratic dependence. The the proposed module doesn't alleviate this. Overall, I do think that the construction does make sense, but the framing is questionable. The approach alleviates the need to process the 5D cost volume _with 5D convolutions_, but it does not alleviate the need to construct the cost volume or to process it. The technical contribution is small, but it does seem effective. I do believe that the paper could be significantly improved by addressing these points. I would like to see an answer by the authors before recommending clear acceptance.

Correctness: The approach and paper seem correct.

Clarity: The paper is well-written and clear. I would try to stronger highlight the fine distinction between learning a feature that is compared using a fixed cost and learning a feature that is compared with a completely learned cost. I would also stronger emphasize the role of the 2D convolutions as compared to the "per-displacement" MLP that is ablated in Table 3. These points are IMHO subtle but important and could be easily missed.

Relation to Prior Work: Relevant papers have been cited. Some relevant work are not included in Table 2 (ScopeFlow [2], ARFlow [A]). Why were these ommited? [A] Liang Liu, Jiangning Zhang, Ruifei He, Yong Liu, Yabiao Wang, Ying Tai, et al. Learning by Analogy: Reliable Supervision from Transformations for Unsupervised Optical Flow Estimation. CVPR 2020.

Reproducibility: Yes

Additional Feedback: --- POST REBUTTAL --- The authors addressed my main concerns in their rebuttal. Additionally, there is a consensus among reviewers that this is a valuable contribution. I'm raising my score accordingly.


Review 3

Summary and Contributions: The paper proposes using a neural network to learning the matching cost between pixel pairs for optical flow. Instead of using operations such as dot product, or cosine similiarity, which decimate the feature dimension, the authors use feature concatentation and convolution which are displacement invariant. The method outperforms published results and all coarse-to-fine flow networks.

Strengths: * The idea of using a learned matching cost makes sense, Table 3 verifies the benefit of using the learning matching cost * The method is simple and faster than closely related works such as VCN * The paper outperforms all pyramid based flow networks on Sintel and is close to the best on KITTI * There are clear quantitative and qualitative support for using the MAP layer * The approach is simple and can be easily integrated into other pyramid based flow networks

Weaknesses: * There are very few ablation experiments to provide insight into the method. For example, Table 3 only includes results on Chairs, while ablations on Sintel and KITTI would be more informative. * I don't think the Memory usage in Table 1 is correct. To my understanding, memory usage should be the same as 4d conv and VCN. Unless they are processing each pixel in the UxV sequentially, which I doubt is what is used in practice, and either way, during training the full KxHxWxUxV grid will need to be stored for backprop. * The method seem more resource intensive than other flow networks, the training schedule is longer than VCN and requires more GPUs

Correctness: The paper follows standard training and evaluation methodology, besides the memory usage in Table 1, all claims appear correct.

Clarity: The paper is well written and easy to follow. However, I had some difficulty understanding the MAP estimation section (MAP is probably not the best acronym to use). It wasn't immediately clear to me that w(u,v) was being learned. The wording "where w(u,v) denotes the correlation between the displacement u and v" was ambiguous

Relation to Prior Work: Due to the strong connection with VCN, I felt that VCN should be discussed further in the related work section. Particularly since both methods use the soft-argmax operator to regress flow, and also the networks have a similar structure. " VCN [26] 77 further proposes to construct multi-channel cost volumes and use separable volumetric filtering" and "VCN [26] leverages an intermediate strategy by concatenating multiple handcrafted matching costs to form a multi-channel cost volume.". I felt that this was not a complete summary of VCN and could better contrast the methods. Also, I think the term "hand-crafted" metric is a bit misleading, because previous flow metrics use the dot product or cosine similiary over learned features, while handcrafted is generally used to refer to metrics such as NCC, SAD, or SSE directly over pixels or grids.

Reproducibility: Yes

Additional Feedback:


Review 4

Summary and Contributions: The authors propose to decouple the 2D displacement of optical flow in constructing the matching cost volume, which can relieve the computation and memory requirements of the method. Extensive experiments and analysis demonstrate its effectiveness and efficiency.

Strengths: + easy to read and follow. + simple idea and it works. + extensive experiments.

Weaknesses: - Table 1 gives some theorical ananlysis of the parameters and memory requirement of different methods. If some real resource usages can be reported, it will be more intersting. - The title of the paper is not very good. "Optical flow by learning matching cost" is not new, in my understanding. Many methods estimate the flow by learning matching cost. Furthermore, the core idea "decoupling" is not reflected in the title at all.

Correctness: The claims and backups are very clear and I think is correct technically, while the title may needs to change.

Clarity: Yes, great writing.

Relation to Prior Work: Yes, clearly presented.

Reproducibility: Yes

Additional Feedback: --- POST REBUTTAL --- The authors discussed the memory usage and resource intensive further in the rebuttal, which fully address my concerns. After reading other reviewers comment, I keep my rating, and recommend a acception.

[Author Response · NeurIPS 2020]

We thank all reviewers for their comments and suggestions. The reviewers have acknowledged that the method is simple and effective; it has demonstrated superior performance on standard benchmarks and 'will have an impact for the general case of computing cost volumes'.

**R1: Q1. DICL versus Reduced DICL:**

To respond to R1's comment, we conducted additional experiments to compare the original DICL with the reduced DICL. Results are given in Table 1. It can be seen that the reduced DICL results in slightly improved performance than the MLP (1.72 *vs* 1.76 on the Chairs dataset), but still has a large gap with the original DICL (1.72 *vs* 1.33).

**R1: Q2. Image guided MAP layer:** We have also tested the image guided MAP layer in our network but only achieved minor performance improvement, *e.g.* less than 0.02 pixel in EPE on the Chairs dataset. Therefore we remove the image guidance in the MAP layer.

Table 1: **Ablation study on cost computation metrics.** The models for 'Chair' were trained on the Chairs dataset. The models for 'K-15' and 'S' were trained on Things dataset.

| Method | Chair | K-15 train | | S-train (EPE) | |
|---|---|---|---|---|---|
| | EPE | EPE | Fl-all | Clean | Final |
| Dot Product | 1.86 | 10.39 | 31.1 | 2.57 | 4.06 |
| Cosine Simi | 1.84 | 10.45 | 30.2 | 2.55 | 4.03 |
| 3-Layer MLP | 1.76 | 9.83 | 28.9 | 2.45 | 3.98 |
| Reduced DICL | 1.72 | 9.77 | 28.3 | 2.42 | 3.99 |
| DICL | 1.33 | 8.78 | 23.8 | 2.11 | 3.85 |

**R1: Q3. Minor corrections:** We will rephrase the words as R1 suggested, including line 156, line 139, and a different acronym for the cost re-weighting process, *e.g.* Displacement-Aware Projection (DAP) layer.

**R2: Q1. Apply DICL to other existing pipelines:**

We replace the non-learned metrics of two well-known pipelines *i.e.* PWCNet and VCN with our DICL module and report the results on the Chairs dataset in Table 2. With our DICL module, both PWCNet and VCN achieve a notable improvement: 8.5% for PWCNet (2.00 *vs* 1.83) and 13.7% for VCN (1.68 *vs* 1.45).

Table 2: **PWCNet and VCN with our DICL module.** Models were trained and evaluated on the Chairs dataset.

| Method | PWCNet | PWCNet + DICL | VCN | VCN + DICL |
|---|---|---|---|---|
| Chair EPE | 2.00 | 1.83 | 1.68 | 1.45 |

**R2: Q2. Is 5D processing a problem?** One of the largest challenge for the optical flow problem is the large search space, although it has been largely alleviated by the coarse-to-fine techniques. Unlike stereo matching that a disparity is always positive, a displacement in optical flow can be either negative or positive. Therefore, when setting the max displacement to 3 on each scale, the corresponding searching window is $7 \times 7$, which has already matched the searching range of deep stereo matching methods (48 for PSMNet on quarter resolution). Moreover, since 4D convolutions will occupy much more GPU memories than 3D convolutions used in stereo, solving optical flow with 5D feature volumes and 4D convolutions is impractical. Similar to deep stereo matching, the cost volume plays a crucial role in ensuring the network to learn matching rather than context-flow mapping. Therefore, we keep the cost volume in our network.

**R2: Q3. Relevant papers:** These are a few related CVPR 2020 papers that were officially published after the deadline of NeurIPS. Upon the reviewer's request, we will include those papers in a revised version for the sake of completeness.

**R4: Q1. Ablation study on Sintel and KITTI:** Per R4's comment, we perform an extra ablation study of our method on the Sintel and KITTI 2015 datasets. As provided in Table 1, our DICL module performs consistently better than other cost computation variants with a large margin.

**R4: Q2. Memory usage and resource intensive:** We agree that, in the training phase the gradients need to be stored in full $K \times H \times W \times U \times V$ grid, but our method needs not to store the full feature volume. We will clarify this part in Table 1 of the paper. Also, as R5 suggested, we will replace the theoretical memory consumption with the actual memory usage. Compared with VCN, our method requires slightly more iterations (150K *vs* 140K) to train on the Chairs dataset, but much faster in inference (0.08s *vs* 0.18s). It is worth noting that our training iterations are much fewer than PWCNet (150K *vs* 1200K).

**R4: Q3. Minor corrections:** We will add a further discussion of our method versus VCN, change the term 'hand-crafted' metrics to 'non-learned', and tighten up language as suggested.

**R5: Q1. Real resource usages:** Upon the reviewer's suggestion, we will replace the theoretical resource usage with the real one, *e.g.* training with a crop size of [256, 384] on the Chairs dataset, it requires 1.9G memory for VCN and 1.1G (58% of the former) for ours to process a pair of images.

**R5: Q2. Revisit title:** Thanks for the suggestion, we may change the title to 'Displacement-invariant matching cost learning for accurate optical flow estimation'.

[Meta-Review · NeurIPS 2020]

Reviewers like the proposed method, which is simple and effective, and the good empirical results. All reviewers recommend acceptance after the rebuttal. AC agrees with this consensus recommendation. The authors should incorporate the reviewers' suggestions, in particular, those regarding clarity, proper discussion of related work (especially VCN), and ablations on Sintel and KITTI.